

**Evaluation of Anthropogenic Emissions and Ozone Pollution in the North China Plain: Insights from the Air Chemistry Research in Asia (ARIAs) Campaign**

Hao He[1], Xinrong Ren[1,2,3], Sarah E. Benish[1], Zhanqing Li[1,4,5], Fei Wang[5,6], Yuying Wang[5], Timothy P. Canty[1], Xiaobo Dong[7], Feng Lv[7], Yongtao Hu[8], Tong Zhu[9], and Russell R. Dickerson,[1,4]

[1]Department of Atmospheric and Oceanic Science, University of Maryland, College Park, MD 20742, USA

[2]Air Resources Laboratory, National Oceanic and Atmospheric Administration, College Park, MD 20742, USA

[3]Cooperative Institute for Climate and Satellites, University of Maryland, College Park, Maryland, USA

[4]Earth System Science Interdisciplinary Center, University of Maryland, College Park, MD 20740, USA

[5]College of Global Change and Earth System Science, Beijing Normal University, Beijing, 100875, China

[6]Key Laboratory for Cloud Physics, Chinese Academy of Meteorological Sciences, Beijing, 100081, China

[7]Weather Modification Office of Hebei Province, Shijiazhuang, 050021, China

[8]School of Civil & Environmental Engineering, Georgia Institute of Technology, Atlanta, GA 30332, USA

[9]College of Environmental Sciences and Engineering, Peking University, Beijing 100871, China

**Keywords:** Ozone Production Sensitivity, Airborne Measurements, OMI, CMAQ

Corresponding to Dr. Hao He (haohe@umd.edu)



## Abstract

To study the air pollution in the North China Plain (NCP), the Air Chemistry Research in
Asia (ARIAs) campaign conducted airborne measurements of air pollutants including $O_3$, CO,
NO and $NO_2$ in spring 2016. High concentrations of pollutants, >100 ppbv of $O_3$, >500 ppbv of
CO, and >10 ppbv of $NO_2$, were observed throughout the boundary layer during the campaign.
CMAQ simulations with the 2010 EDGAR emissions can capture the basic spatial and temporal
variations of ozone and its major precursors such as CO, $NO_x$ and VOCs, but significantly
underestimate their concentrations. Observed emission enhancements of CO and $NO_x$ with
respect to $CO_2$ suggest the existence of combustion with high emissions such as biomass burning
in the NCP. The comparison with emission factors from the 2010 EDGAR emission inventory
indicates that the contribution of combustion with high emissions has been overestimated.
Differences between CMAQ simulations with 2010 emissions and satellite observations in 2016
can reflect the change in anthropogenic emissions. $NO_x$ emissions decreased in megacities such
as Beijing and Shanghai confirming the effectiveness of recent control measures in China, while
in other cities and rural areas $NO_x$ emissions slightly increased, e.g., CMAQ predicts only ~80%
of $NO_x$ observed in the aircraft campaign area. CMAQ also underestimates HCHO (a proxy of
VOCs, by ~20%) and CO (by ~60%) over the NCP, suggesting adjustments of the 2010 EDGAR
emissions are needed to improve the model performance. $HCHO/NO_2$ column ratios derived
from OMI measurements and CMAQ simulations show that VOC-sensitive chemistry dominates
the ozone photochemical production in eastern China, suggesting the importance of tightening
regulations on VOCs emissions. We adjusted EDGAR emissions based on satellite observations,
conducted sensitivity experiments of CMAQ, and achieved better model performance in
simulating ozone, but underestimation still exists. Because of the VOC-sensitive environment in
ozone chemistry over the NCP, future study and regulations should focus on VOCs emissions
with the continuous controls on $NO_x$ emissions in China.





## 1. Introduction

With rapid economic growth in the past three decades, the consumption of energy in
China increased dramatically (Zhang and Cheng, 2009; Guan et al., 2018; Shan et al., 2018).
Fossil fuels dominate total energy consumption, with coal still accounting for more than 50% of
the carbon dioxide ($CO_2$) emissions in China (Shan et al., 2018). This drastic increase in fossil
fuel energy consumption is accompanied with deterioration of air quality (Chan and Yao, 2008;
Fang et al., 2009), posing a threat to public health (Tie et al., 2009; Kan et al., 2012; Chen et al.,
2013; Lelieveld et al., 2015). Particulate matter (PM) pollution, especially $PM_{2.5}$ in the North
China Plain (NCP), drew public concern and governmental actions (He et al., 2001; Ye et al.,
2003; Wang et al., 2005; Sun et al., 2006; Yang et al., 2011; Zhang et al., 2012; Zhang et al.,
2013). PM pollution also has complex interactions with the planetary boundary layer (PBL) and
its evolution, which can further degrade the air quality (Guo et al., 2016; Li et al., 2017b). Recent
studies showed that tropospheric ozone ($O_3$) pollution increased in China which exacerbated its
complex air pollution problem (Xue et al., 2014; Verstraeten et al., 2015; Wang et al., 2017b; Ni
et al., 2018).
Elevated ozone concentrations have adverse impacts on both human health (WHO, 2003;
Anderson, 2009; Jerrett et al., 2009) and the ecosystem (Adams et al., 1989; Chameides et al.,
1999; Ashmore, 2005). Tropospheric ozone absorbs thermal radiation and acts as the third most
important anthropogenic contribution to radiative forcing of climate (Ramanathan and
Dickinson, 1979; Lacis et al., 1990; IPCC, 2014). In the lower troposphere, the photolysis of
ozone is an important source of atmospheric hydroxyl (OH) radicals that control the lifetimes of
atmospheric species such as CO and volatile organic compounds (VOCs) (Logan et al., 1981;
Thompson, 1992; Finlayson-Pitts and Pitts, 1999). Tropospheric ozone has a relatively long
lifetime of several days to weeks (Stevenson et al., 2006; Young et al., 2013), leading to
significant long-range transport of ozone and its precursors (Jacob et al., 1999; Derwent et al.,
2004; Lin et al., 2008). Thus, investigation of ozone pollution in China is essential to support the
national and international policy decision for air quality and the climate.
Tropospheric ozone is produced through complex photochemical reactions of precursors
including nitrogen oxides ($NO_x = NO + NO_2$) and VOCs in the presence of sunlight
(Haagensmit, 1952; Crutzen, 1974; Fishman et al., 1979; Seinfeld and Pandis, 2006). In China,
sectors of power generation, industry, and transportation dominates the $NO_x$ emissions (Streets et



al., 2003; Ohara et al., 2007; Zhao et al., 2013a). Before 2010, $NO_x$ emissions in China increased substantially (Lin et al., 2010a; Zhao et al., 2013c). Analysis of satellite data revealed that recently $NO_x$ emissions have started decreasing in highly developed regions such as the Pearl River Delta (PRD), but still increased in other regions (Gu et al., 2013; Duncan et al., 2016; Liu et al., 2016). Anthropogenic VOCs emissions had a similar increasing trend in the past decades (Bo et al., 2008; Wei et al., 2011; Kurokawa et al., 2013; Zhao et al., 2017) and are projected to increase in the future (Zhang et al., 2018). Therefore the recent increase of tropospheric ozone in China could likely be explained by the enhanced anthropogenic emissions of ozone precursors.

Due to the complex $O_3$-$NO_x$-VOCs chemistry, we need to investigate the photochemical regime for local ozone production, i.e., $NO_x$-sensitive or VOC-sensitive (Dodge, 1987; Kleinman, 1994). Duncan et al. (2010) used the ratio of tropospheric columns of formaldehyde (HCHO) and nitrogen dioxide ($NO_2$) observed by the National Aeronautics and Space Administration (NASA) Aura Ozone Monitoring Instrument (OMI) to characterize ozone sensitivity. Studies show that a $NO_x$-sensitive regime dominates in the United States, except in megacities such as Los Angles and New York City where the local ozone production is in VOC-sensitive or in transition regimes (Duncan et al., 2010; Jin et al., 2017; Ring et al., 2018). However, VOC-sensitive and transition regimes for ozone photochemical production exist ubiquitously in China due to large amount of $NO_x$ emissions, especially over the NCP (Chou et al., 2009; Xing et al., 2011; Jin and Holloway, 2015; Jin et al., 2017). As such, although the current regulations in China focus only on reduction of $NO_x$ emissions (Wang and Hao, 2012; Wang et al., 2014a), air quality might also benefit from VOCs controls.

Aircraft measurements are essential to study the precursor emissions, photochemical production, and transport of ozone pollution at regional scale. However, airborne campaigns are sparse in China (Dickerson et al., 2007; Zhang et al., 2014; Ding et al., 2015; Huang et al., 2015; Wang et al., 2017a). To better understand the characteristics of ozone pollution, the Air Chemistry Research in Asia (ARIAs) aircraft campaign was conducted in Hebei Province of the NCP during May-June 2016, which was affiliated with the surface Aerosol Atmosphere Boundary-Layer Cloud ($A^2BC$) experiment (Wang et al., 2018a; Wang et al., 2018b). Concentrations of major air pollutants in the lower atmosphere were measured during 11 research flights in the NCP, which were conducted in association with the NASA Korea U.S. – Air Quality (KORUS-AQ) campaign in downwind South Korea. Measurements collected in the



ARIAs research flights and the A²BC surface observations provide a comprehensive dataset to
thoroughly study the tropospheric ozone pollution and emissions of its precursors in China.
In this study, we evaluate anthropogenic emissions and the ozone pollution in the NCP
using a combination of aircraft measurements, surface monitoring data, satellite observations,
and modeling results. The Environmental Protection Agency (EPA) Community Multiscale Air
Quality (CMAQ) model was used to simulate the atmospheric chemistry for the ARIAs
campaign. We evaluate the emission data by comparing with the aircraft measurements and
satellite products, and then adjust emissions to improve the CMAQ performance. Lastly, we
investigate the sensitivity of ozone production derived from CMAQ simulations and OMI
observations and discuss the future ozone pollution in China.

**2. Data and Method**
**2.1 Aircraft campaign in the NCP**
With more than 250 million tons of iron and steel produced in 2016 (data from
http://data.stats.gov.cn, accessed in September 2018), Hebei Province in the NCP is the most
industrialized area in China. Due to its high emissions and proximity to megacities Beijing and
Tianjin, the Beijing-Tianjin-Hebei area has experienced severe air pollution in the past decade
(Zhao et al., 2013b; Wang et al., 2014b). In May and June 2016, the ARIAs aircraft campaign
was conducted over Hebei Province to investigate the emissions, chemical evolution, and
transport of air pollutants. The airborne campaign was coordinated with the A²BC field campaign
in Xingtai (XT, 37.18 °N, 114.36 °E, 182 m above sea level, ASL) and the NASA KORUS-AQ
campaign to expand the study to East Asia. A Harbin Y12 research airplane (similar to the de
Havilland Twin Otter) was employed to measure concentrations of air pollutants including $O_3$,
carbon monoxide (CO), $CO_2$, $NO_2$, and aerosol optical properties. The research airplane was
located in Luancheng airport, hereafter referred to as LC (LC, 37.91 °N, 114.59 °E, 58 m ASL),
south of Shijiazhuang, the capital city of Hebei province with a population of 10 million. Eleven
research flights were conducted during the ARIAs campaign (Fig. 1a). Vertical profiles of air
pollutants from near surface (~100 m above ground level, AGL) to the free troposphere (> 3000
m) were conducted over LC, XT (the supersite of the A²BC campaign), Julu (JL, 37.22 °N,
115.02 °E, 30 m ASL), and Quzhou (QZ, 36.76 °N, 114.96 °E, 40 m ASL).
The airborne measurements of ozone were conducted using a commercially available





analyzer (Model 49C, Thermo Environmental Instruments, TEI, Franklin, Massachusetts)
(Taubman et al., 2006). $NO_2$ was measured using a modified commercially available cavity ring-
down spectroscopy (CRDS) detector (Castellanos et al., 2009; Brent et al., 2013). Concentrations
of CO and $CO_2$ were monitored with a 4-channel Picarro CRDS instrument (Model G2401-m,
Picarro Inc., Santa Clara, CA), calibrated with $CO/CO_2$ standards certified at the National
Institute of Standards and Technology (Ren et al., 2018). All the instruments were routinely
serviced, calibrated and used for airborne measurements in the United States and China
(Taubman et al., 2006; Dickerson et al., 2007; Hains et al., 2008; He et al., 2012; He et al., 2014;
Ren et al., 2018; Salmon et al., 2018). Measurements of ambient air pollutants were made at 1
Hz frequency and synchronized with time, geolocation and altitude from the Global Position
System (GPS).

In the ARIAs research flights, 28 whole air samples (WAS) were collected in vertical

spirals at different altitudes from ~400 m to ~3500 m. The WAS were analyzed using gas
chromatography (GC) with Flame Ionization Detection (FID) and Mass Spectroscopy (MS) by
the College of Environmental Sciences and Engineering at Peking University. 74 species of
alkanes, alkenes/alkynes, aromatics, and halocarbons were identified and quantified for a study
on ozone photochemical chemistry (see details in Benish et al., 2019). Detection limits for the
compounds ranged from 2 to 50 pptv. Surface observation of trace gases including $O_3$, CO, NO,
and $NO_x$ were measured at the $A^2BC$ Xingtai supersite using analyzers manufactured by Ecotech
(Wang et al., 2018b); detailed description of the analyzers is discussed in Zhu et al. (2016).
Surface HCHO concentrations were monitored using a formaldehyde analyzer (AERO LASER,
Germany, Model 4021) based on fluorometric Hantzsch reactions (Gilpin et al., 1997;
Rappenglück et al., 2010). All surface observations were collected as 1-min averaged data and
processed to hourly mean values.

**2.2 Satellite products**

To evaluate the emissions and atmospheric chemistry in the NCP and greater East Asia,

we used satellite observations of CO, $NO_2$, and HCHO for May and June 2016. The
Measurements of Pollution In the Troposphere (MOPITT) instrument onboard the NASA Terra
satellite retrieved CO column contents with ~10:30 am local overpass time (Deeter et al., 2003).
We used the latest version 7 MOPITT Level 3 daily gridded average products ($1° \times 1°$ spatial





resolution, available at https://eosweb.larc.nasa.gov/project/mopitt/mop03j_v007) for the ARIAs
campaign period (MOPITT Science Team, 2013). MOPITT thermal-infrared and near-infrared
(TIR + NIR) products shows improved sensitivity to near surface CO in China (Worden et al.,
2010). We used MOPITT near surface CO (~ 900 hPa) products and related averaging kernels
(AKs) to evaluate the CMAQ results (Deeter et al., 2012).
OMI, onboard the NASA Aura satellite, is a UV/Vis solar backscatter spectrometer in a
polar sun-synchronous orbit with a ~1:35 pm local overpass time. With high spatial resolution
(13 km × 24 km for the center at nadir) and nearly daily coverage, OMI has provided monitoring
of trace gases and aerosol properties since 2005 (Levelt et al., 2006). The Version 3 OMI Level 2
$NO_2$ products (https://disc.gsfc.nasa.gov/datasets/OMNO2_V003/summary) (Krotkov et al.,
2018) were used to evaluate the emissions and atmospheric chemistry in East Asia. Under clear
sky, tropospheric $NO_2$ columns from OMI has precision of ~0.5 × $10^{16}$ molecules cm$^{-2}$ and an
accuracy of ±30% (Krotkov et al., 2017). OMI HCHO Version 3 Smithsonian Astronomical
Observatory (SAO) (https://disc.gsfc.nasa.gov/datasets/OMHCHO_V003/summary) Level 2
products were used in this study (Chance, 2007; González Abad et al., 2015). The precision of
column HCHO is ~1.0 × $10^{16}$ molecules cm$^{-2}$ and SAO products have an accuracy of ±25-30%
without cloud (Millet et al., 2006; Boeke et al., 2011). Data in OMI pixels affected by the row
anomaly and contaminated by clouds were filtered out using quality flags for both $NO_2$ and
HCHO columns.

**2.3 Model set-up**
We used CMAQ version 5.2 (EPA, 2017) to simulate atmospheric chemistry for the
ARIAs campaign. The Weather Research and Forecasting (WRF) model Version 3.8.1
(Skamarock et al., 2008) was driven by the European Centre for Medium-Range Weather
Forecasts (ECMWF) ERA-Interim products (ds627.0, https://rda.ucar.edu/datasets/ds627.0) (Dee
et al., 2011) to simulate meteorological fields. Two domains with spatial resolution of 36 km and
12 km (Fig. 1b) were used to cover East Asia, with 35 layers from the surface to 50 hPa and ~20
layers in the lowest 2 km. Major physical options in WRF include the Rapid Radiative Transfer
Model (RRTM) radiation scheme (Clough et al., 2005), the Pleim-Xiu surface layer and land
surface model (Pleim and Xiu, 1995; Xiu and Pleim, 2001), the Asymmetric Convective Model
(ACM2) boundary layer scheme (Pleim, 2007), the Kain-Fritsch cumulus scheme (Kain, 2004),




and the WRF Single-Moment 6 (WSM-6) microphysics (Hong and Lim, 2006). The National

Centers for Environmental Prediction (NCEP) ADP Global Surface and Upper Air Observational

Weather Data (ds461.0 and ds351.0, https://rda.ucar.edu) were used to perform observational and

analysis nudging on all domains following the method developed for NASA aircraft campaigns

(He et al., 2014; Mazzuca et al., 2016). WRF outputs were processed by the EPA Meteorology-

Chemistry Interface Processor Version 4.3 (MCIP v4.3, released in November 2015) for emission

processing and CMAQ simulations.

Anthropogenic emissions were from the Emissions Database for Global Atmospheric

Research Version 4.2 (EDGAR v4.2, 0.1° × 0.1° resolution) of year 2010, which are widely used

for chemical transport modeling (European Commission, 2011). We used the EPA Sparse Matrix

Operator Kernel Emissions (SMOKE) modeling system Version 4.5 (UNC, 2017) to project

EDGAR emissions to the modeling domain. Emissions of air pollutants were speciated into

Carbon Bond 05 chemical mechanism (Yarwood et al., 2005) and updated AERO6 aerosol

module (Appel et al., 2013). The EDGAR v4.2 inventory has emissions for energy, industry,

residential, and transport sectors. Without stack height information for power plants in the energy

sector, we followed the approach developed in He et al. (2012) to locate these anthropogenic

emissions at ~200 m above the surface as an approximation for averaged stack height and plume

rise. We used the United States Geological Survey (USGS) 24 category land use dataset

combined with the Biogenic Emission Inventory System (BEIS) emission factors table to

generate the input files for the CMAQ inline biogenic emissions modeling. Biogenic emissions

were estimated using the BEIS module inline in CMAQ (EPA, 2017).

CMAQ v5.2 uses the updated Carbon Bond 6 (CB6r3) chemical mechanism (Yarwood et

al., 2010) including improved chemistry mechanism for organic nitrates and peroxyacyl nitrates

(PAN) chemistry and will lead to better performance for simulating Secondary Organic Aerosols

(SOA) and tropospheric ozone in the United States (Appel et al., 2016). CMAQ was run with a

coarse domain and a nested domain (Fig. 1b). Chemical initial and boundary conditions for the

coarse domain were obtained from the default concentration profiles built in CMAQ (EPA,

2017). Results from the CMAQ coarse domain were used to generate boundary conditions for

the nested domain. The WRF-CMAQ system was run from mid-April to June with the first 2

weeks as spin-up. Hourly concentrations of air pollutants were saved for further analysis and

model evaluation.





## 3. Results and discussion

### 3.1 Air Pollution in the NCP and CMAQ performance

Figure 2 summarizes all aircraft measurements of $O_3$, $NO_2$, CO, and $CO_2$ over the NCP from eleven research flights. Generally, we observed high concentrations of trace gases, such as >100 part per billion by volume (ppbv) of $O_3$, >20 ppbv of $NO_2$, >500 ppbv of CO, and >450 part per million by volume (ppmv) of $CO_2$, in the aircraft campaign area (defined as 36.5-38.5°N, 114.0-115.5°E hereafter). We conducted vertical spirals over XT (the $A^2BC$ supersite), LC (the airport in south of Shijiazhuang), and two rural areas (JL and QZ) during the ARIAs research flights. Figure 3a summarizes vertical distributions and the mean profiles of air pollutants over XT, with mean $O_3$ concentrations of 80 ppbv in the lower atmosphere. We observed isolated plumes with >10 ppbv of $NO_2$, >1000 ppbv of CO, and >440 ppmv of $CO_2$ over XT, usually with a secondary maximum between 800 and 1200 m. These plumes aloft can play an important role in long-range transport of air pollutants to downwind regions. Profiles over LC (Fig. 3b) show higher $O_3$ concentrations (>100 ppbv) and relatively moderate $NO_2$ (~3 ppbv) and CO (~250 ppbv). The rural areas, JL and QZ, have relatively clean environment with <80 ppbv of $O_3$, <2 ppbv of $NO_2$, and <300 ppbv of CO (Fig. 3c and 3d). Even the concentrations of air pollutants over the rural region in the NCP are comparable or higher than values in urban areas in North America and Europe. In summary, we found the south-north and east-west gradients of air pollution, i.e., higher concentrations of air pollutants in the west XT-LC corridor near the mountain as compared with east side of JL and QZ. Thus, the ARIAs research flights have good coverage of regions with both high and moderate concentrations of air pollutants and can fairly represent the regional nature of air pollution over the NCP.

Comparison of the surface trace gas observations at the Xingtai supersite and the CMAQ simulations driven by the EDGAR inventory (named baseline CMAQ case hereafter) reveals that CMAQ generally underestimates concentrations of major air pollutants (Fig. S1 in the supplementary material). The baseline CMAQ run successfully captures the diurnal and daily variations of surface ozone in Xingtai, although consistently underpredicts its concentrations. For CO and $NO_x$, two important ozone precursors, CMAQ substantially underestimates their concentrations in Xingtai by more than 50% and especially fails to capture the extremely high values such as 6~7 ppmv of CO and ~100 ppbv of $NO_x$. This underestimation could be caused by local sources poorly represented in the 12-km model simulations. For ambient HCHO, an



important byproduct of VOC oxidization in ozone photochemical production, the baseline CMAQ run captures the variations, but substantially underestimates its concentrations. These results suggest that the underestimation of ozone precursors in CMAQ could lead to the poor model performance of simulating tropospheric ozone and other pollutants.

Similar analyses were conducted to investigate air pollutant concentrations in the lower troposphere over the NCP observed by the aircraft. A case of the research flight on June 11, 2016 (Fig. S2 in the supplementary material) shows that CMAQ well captures the vertical gradient of air pollutants, while substantially underestimates concentrations of all trace gases except $NO_y$. Following the approach described in Goldberg et al. (2016), we calculated the 10-min average $O_3$, CO, NO, and $NO_2$ concentrations from aircraft measurements and compared them with the baseline CMAQ simulations (Fig. 4) and found similar underestimation (50% to 75% for all air pollutants) as compared with surface measurements (Fig. S1 in the supplementary material). CMAQ overestimates $NO_y$ but substantially underestimates NO and $NO_2$, which suggests that a significant amount of reactive nitrogen compounds could exist in the format of organic nitrates or nitrate aerosols in the model. Figure 5 compares total VOCs concentrations from WAS samples and CMAQ simulations, indicating that VOCs levels are significantly underestimated by 80%. The model evaluation with surface and aircraft measurements suggest that in the NCP ozone pollution has been significantly underestimated in the baseline CMAQ run, which could be due to the uncertainty introduced by using the 2010 EDGAR emissions to simulate the 2016 ARIAs campaign period. Thus, we need to evaluate the emissions inventory data to improve the CMAQ performance and investigate the sensitivity of ozone production.

**3.2 Evaluation of emissions inventory in the NCP**

The EDGAR v4.2 emission inventory in East Asia was created based on the 2010 MIX emission inventory (Li et al., 2017a), so substantial changes were anticipated when used for the ARIAs campaign in 2016. Anthropogenic emission inventories are usually based on the "bottom-up" approach, which relies on the statistics of fossil fuel usage and emission factors (EFs) for each sector defined as the ratio of the amount of air pollutants released by a unit of $CO_2$ emissions, e.g. $CO/CO_2$ and $NO_x/CO_2$. To evaluate the emission inventory data in the NCP, we used a 60-s moving window and conducted linear regression of observed air pollutant (CO, $NO_x$, etc.) concentrations vs. $CO_2$ concentrations, i.e. $\Delta CO/\Delta CO_2$ and $\Delta NO_x/\Delta CO_2$, defined as



emission enhancements (EEs). Through only selecting EEs that are in the PBL (below 1.5 km
AGL in this study) and statistically significant ($R^2 > 0.6$), the values of EEs can act as a proxy of
EFs in the air mass observed (Halliday et al., 2018).
EEs observed during the research flights have a broad range of values. $\Delta CO/\Delta CO_2$ ranges
from below 1%, a typical value of modern automobile emissions, to higher than 10%, a value
indicating fossil fuel combustion with high emissions such as biomass burning (Fig. 6a and 6b).
The mean of observed EE for CO (3.7%) is close to that calculated from the EDGAR inventory
(4.0%) in the aircraft campaign area. Observed $\Delta NO_x/\Delta CO_2$ ratios also have isolated high values
(>0.1%) with a mean value of 0.05%, which is substantially higher than the EF (~0.03%) derived
from the EDGAR inventory. Since estimation of anthropogenic $CO_2$ flux in an urban/suburban
area is challenging (Cambaliza et al., 2014; Heimburger et al., 2017), the underestimation of CO
and $NO_x$ in the NCP could be caused by either underestimated EFs or uncertainty in
anthropogenic $CO_2$ emission data used in the 'bottom-up' approach.
To further investigate the characteristics of air pollutant emissions in the NCP, we
conducted a similar analysis of $\Delta NO_x/\Delta CO$, which are usually co-emitted in combustion
processes. Since around half of the CO and $NO_x$ are from mobile sources in the EDGAR
emission inventory, this ratio can approximately represent the emission characteristic of mobile
sources in the NCP. The mean observed $\Delta NO_x/\Delta CO$ ratio is ~1.3%, significantly lower than
5.6% based on the EDGAR emission inventory (Fig. 6c). These results suggest that the EDGAR
emission inventory substantially overestimates the ratios of $NO_x/CO$, while the automobile
emissions over the NCP in 2016 have been greatly improved due to recent regulations focusing
on $NO_x$, i.e., EDGAR overestimates the contribution from combustion with high emissions. It is
worth noting that we only evaluated the emission ratios (EEs or EFs) in the EDGAR inventory,
while the underestimation of CO and $NO_x$ emissions could be caused by inaccurate $CO_2$
emissions which have not been examined in this study.

**3.3 Evaluation of CO, $NO_x$, and VOCs emissions using satellite data**
Satellite observations are widely used to evaluate the anthropogenic emissions in East
Asia sometimes supplemented by model simulations, e.g., CO emissions using the MOPITT CO
products (Jiang et al., 2015; Zheng et al., 2018), anthropogenic $NO_x$ emissions using OMI $NO_2$
products (Wang et al., 2012; de Foy et al., 2015; Qu et al., 2017), and VOCs emissions using



OMI HCHO products (Stavrakou et al., 2016). In this study, we used measurements from
multiple satellite instruments to evaluate the CMAQ performance of $NO_2$, HCHO, and CO. Since
$NO_2$ and HCHO can be treated as proxy of $NO_x$ and VOCs emissions, we can further improve
the 2010 EDGAR emissions over the NCP base on satellite data.

We followed the approach developed in Canty et al. (2015) to compare the tropospheric

column contents of $NO_2$ from OMI products and CMAQ simulations. Level 2 OMI $NO_2$ swatch
information including row anomaly and quality flags were used to sample $NO_2$ vertical profiles
from CMAQ outputs, and then CMAQ $NO_2$ column was calculated using the OMI averaging
kernel (AK). Lastly, we averaged OMI and CMAQ $NO_2$ column contents to create daily 0.25° ×
0.25° Level 3 products (see details in Canty et al., 2015). A similar approach was used to
integrate HCHO column contents from CMAQ simulations based on OMI HCHO retrievals (see
details in Ring et al., 2018) and construct daily 0.25° × 0.25° Level 3 HCHO products. For
tropospheric CO, we selected the CO concentrations at ~ 900 hPa in CMAQ and averaged them
to 1.0° × 1.0° daily products using MOPITT CO averaging kernel (MOPITT Science Team,
2013). All gridded daily data of satellite and CMAQ were averaged in May and June 2016 for
comparison.

Figure 7a shows strong signals over the NCP of the OMI $NO_2$ observations. CMAQ

underestimates $NO_2$ columns over the aircraft campaign area, and only predicts 81% of $NO_2$
column as compared with OMI observations. However, in urban regions such as Beijing, the
Yangtze River Delta (YRD), and the PRD, CMAQ substantially overestimates column $NO_2$ by
up to 30%. Because the baseline CMAQ simulations used the 2010 anthropogenic emission data,
these differences should reflect the changes in $NO_x$ emissions due to recent air pollution
regulations. The comparison of $NO_2$ column suggests that $NO_x$ pollution of megacities in China
has been substantially improved after 2010 while $NO_x$ pollution in smaller cities and rural area
has worsened, consistent with results from independent studies using OMI (Duncan et al., 2016;
Krotkov et al., 2016). OMI HCHO retrievals also show high values over the NCP in spring when
plants' photosynthetic activity is relatively low, reflecting that the domination of anthropogenic
VOCs emissions in north China (Zhao et al., 2017). CMAQ has good agreement with OMI
HCHO within the aircraft campaign area (<20% underestimation), but substantially
underestimates HCHO columns in south China where biogenic VOCs dominate (Fig. 7b). The
MOPPIT products show high near-surface CO concentrations over the eastern China (Fig. 7c),





while the baseline CMAQ run substantially underestimates CO concentrations over north China
and only predicts 42% of the CO over the aircraft campaign area.
Using $NO_2$ and HCHO as proxies of $NO_x$ and anthropogenic VOCs emissions, the
comparison of satellite observations and the baseline CMAQ simulations suggests that both $NO_x$
and VOCs emissions in the aircraft campaign area need to be adjusted for a better simulation of
tropospheric ozone. Also, the underestimation of CO, as an important precursor, can lead to
underprediction of tropospheric ozone. We calculated the model/satellite ratios of $NO_x$, HCHO,
and CO in East Asia (Fig. 8) and used these ratios to adjust their anthropogenic emissions in
CMAQ. The results will be discussed in Section 3.4.

**3.3 Tropospheric ozone production sensitivity from OMI and CMAQ**
Photochemical production of tropospheric ozone is highly non-linear and dependent on
concentrations of $NO_x$ and VOCs (Kleinman, 1994; Sillman, 1999; Kleinman, 2000). A
maximum rate of ozone production can be achieved with an optimal VOCs/$NO_x$ ratio. With other
VOCs/$NO_x$ ratios, ozone production can be either in the VOC-sensitive regime (the rate of ozone
production is controlled by VOCs concentrations) or in the $NO_x$-sensitive regime (the rate of
production is controlled by $NO_x$ concentrations). Different pollution control strategies can be
implemented to reduce the tropospheric ozone levels in these two regimes. For instance, in a
VOC-sensitive environment, reducing $NO_x$ emissions will lead to limited effects until the ozone
production has been changed to a $NO_x$-sensitive environment with the continuous removal of
$NO_x$ from the atmosphere. Duncan et al. (2010) developed an approach using OMI HCHO/$NO_2$
column ratio to estimate the ozone production sensitivity as: 1) HCHO/$NO_2$ <1: VOC-sensitive
regime; 2) HCHO/$NO_2$ 1~2: transition regime; 3) HCNO/$NO_2$ > 2: $NO_x$-sensitive regime.
Studies show that urban areas in the U.S. such as Los Angeles, New York City and Houston are
in VOC-sensitive or transition regimes, which lead to difficulty in local regulation of air quality
(Duncan et al., 2010; Mazzuca et al., 2016; Ring et al., 2018). Recent studies suggest new
threshold values of HCHO/$NO_2$ ratios between VOC-sensitive, transition, and $NO_x$-sensitive
regimes in the U.S. (Jin et al., 2017; Schroeder et al., 2017).
Using the Duncan et al. (2010) approach, studies using OMI products suggest large areas
of eastern China are either in VOC-sensitive regime (mostly megacities such as Beijing) or in
transition regime (Jin and Holloway, 2015; Jin et al., 2017; Xing et al., 2018). We follow the



approach described in Ring et al. (2018) to calculate the column $HCHO/NO_2$ ratios from OMI
observations and CMAQ simulations for East Asia. OMI column $HCHO/NO_2$ ratios suggest that
the ozone photochemical production is VOC-sensitive or in transition region over the NCP and
other major urban areas such as YRD and PRD (Fig. 9a) if the Duncan et al. (2010) approach is
applicable for these areas. CMAQ successfully captured the spatial distribution of the regional
nature of ozone production sensitivity in eastern China but predicted that the rate of ozone
production is controlled more by VOCs with the CMAQ $HCHO/NO_2$ ratio lower than 1.0 in
Beijing, YRD, and PRD (Fig. 9b). The VOC-sensitive environment suggests the rate of ozone
photochemical production in the NCP is controlled not only by $NO_x$ emissions, but also by
VOCs emissions which currently lack regulations in China. With continuous reduction of
anthropogenic $NO_x$ emissions in China, VOCs controls might be efficient in these VOC-sensitive
regions.

**3.4 Improvements of tropospheric ozone simulation using satellite products**

Results of the previous two sections show that the baseline CMAQ run substantially

underestimates the concentrations of ozone and its major precursors in the NCP. To identify the
individual and combined effects of the emission discrepancy of impacting major ozone
precursors in the NCP, we designed a series of sensitivity experiments with emissions adjusted to
satellite observations. Unlike the top-down approach using global chemical transport models
such GEOS-Chem (Lin et al., 2010b; Qu et al., 2017), here we simply applied the ratios of air
pollutant column contents from satellite observations and CMAQ simulations on each 0.25
degree grids (Fig. 8) as: $CO_{CMAQ}/CO_{MOPITT}$, $NO_{2CMAQ}/NO_{2OMI}$, and $HCHO_{CMAQ}/HCHO_{OMI}$ ratios
for anthropogenic CO, $NO_x$, and VOCs emissions, respectively. To estimate the contribution
from biogenic VOCs emissions, we conducted one more run with the in-line BEIS module
turned off. Table 1 shows the emission adjustments for the five sensitivity experiments. CMAQ
was run for the nested 12 km domain (D02) with the same meteorology, initial conditions, and
boundary conditions derived from the coarse domain simulations.

Figure 10 presents the evaluation of surface observations with respect to two sensitivity

experiments (CMAQ_baseline and CMAQ_all, comparison with all CMAQ runs are presented in
Fig. S3 in the supplementary material). CMAQ still might not capture the extreme high values of
surface $O_3$ and CO (Fig. 10a and 10b). For instance, the maximum CO concentration from




CMAQ simulations are ~1700 ppbv while surface observations have CO peaks higher than 6000
ppbv (Fig. 10b). The adjustments of the emission inventory have improved the model
simulations of $NO_2$/NO (Fig. 10c and 10d) and HCHO (Fig. 10e). During the ARIAs flights, we
observed various sources of emissions in the aircraft campaign area such as small factories and
biomass burning, which are not included in the EDGAR emission inventory. Thus, the reason for
the model underestimation could be that the spatial resolution (12 km) of the nested CMAQ
domain cannot represent the detailed emissions and resolve the local air pollution hotspots.
However it is worth noting that even our CMAQ system is still not capable to reproduce the
surface air quality at Xingtai, the adjustments of EDGAR emissions based on satellite
observations reduce the underestimation.

The ARIAs flights covered a large area (~$10^4$ km$^2$) in Hebei Province, which represent

the regional nature of air pollution over the NCP. A case comparison of CMAQ_All case and
Y12 measurements on June 11, 2016 (Fig. 11) shows better results in both concentrations and
vertical gradient of air pollutants (compared with Fig. S2 in the supplementary material),
indicating the effectiveness of improving the emission inventories based on satellite
observations. Table 2 summarizes the model performance of CMAQ as compared with aircraft
measurements. The adjustments of the EDGAR emissions with satellite observations moderately
improved simulations of ozone pollution, with the root mean square error (RSME) decreasing
from 25.1 ppbv (the baseline case) to 21.2 ppbv (CMAQ_All case) and the mean ratio of CMAQ
simulations to aircraft observations increasing from 0.75 to 0.82. The model performance of CO
has also been improved, with the RMSE decreasing from 247.0 ppbv to 203.6 ppbv and the mean
ratio increasing from 0.40 to 0.66. For nitrogen compounds including $NO_2$, NO, and $NO_y$, the
adjustments of EDGAR emissions have small impacts on improving the CMAQ performance.
The reason could be that the ozone photochemistry is mainly VOC-sensitive over the NCP, so the
adjustments of $NO_x$ emissions have limited impacts close to sources.

**4. Conclusions and Discussion**

The ARIAs campaign conducted aircraft measurements over the NCP and observed high

concentrations of air pollutants including $O_3$, CO, and $NO_x$. CMAQ simulations driven by the
2010 EDGAR emissions substantially underestimate the levels of ozone and its precursors in the
campaign region. Analysis of emission enhancements of CO and $NO_x$ with respect to concurrent



459 $CO_2$ measurements suggests that the usage of the 2010 EDGAR emissions for the 2016 ARIAs

460 campaign could introduce substantial uncertainty due to the recent changes of anthropogenic

461 emissions in China. Comparison of atmospheric columns of $NO_2$ from CMAQ simulations and

462 satellite observations suggests that $NO_x$ emissions decreased in megacities such as Beijing and

463 Shanghai but increased in rural areas from 2010 to 2016. Similar analysis of HCHO and CO

464 shows that the EDGAR VOCs and CO emissions could also be underestimated in the NCP.

465 HCHO/$NO_2$ column ratio from OMI observations indicates tropospheric ozone production is

466 mainly in the VOC-sensitive regime in the NCP, which is also confirmed by CMAQ simulations.

467 To test a hypothesis that the poor model performance is due to emission biases, we adjusted the

468 EDGAR emissions over East Asia based on satellite observations. Better performance of

469 simulating ozone and its precursors is achieved, while underestimation still exists.

470  Both satellite observations and CMAQ simulations indicate that the VOC-sensitive

471 chemistry dominates the ozone photochemical production in eastern China, so the rate of local

472 ozone production is mainly controlled by the VOCs emissions. In the past few years, despite

473 implementation of control measures mainly on $SO_2$ and $NO_x$, ozone concentrations have

474 increased in China. Our study indicated that high $NO_x$ concentrations were pervasive in the PBL

475 over rural areas of the NCP, where anthropogenic VOCs were also abundant. Reducing $NO_x$

476 emissions is essential to control ozone on the regional scale, but our model simulations indicated

477 that reducing VOCs emissions can lower the rate of photochemical smog production.

478  Currently, studies and regulations on anthropogenic VOCs emissions in China are

479 lacking, so with expectation of further decreasing $NO_x$ emissions in China, more severe ozone

480 pollution could be anticipated. It is worth noting that while VOCs controls can have beneficial

481 impact on the local rate of ozone production in the VOC-sensitive regime, the ozone levels will

482 not decrease until $NO_x$ emissions are substantially lower, i.e., regulations on VOCs are needed as

483 well as the continuous controls on $NO_x$ emissions in China. These results can also partially

484 explain why ozone pollution emerged in the past few years while $PM_{2.5}$ pollution has been

485 substantially improved with strict regulations on anthropogenic emissions. New datasets such as

486 the updated 'bottom-up' emissions inventory for East Asia and high resolution satellite

487 observations such as TROPOMI and GEMS products are needed to improve the modeling of

488 ozone pollution in China, which can provide scientific evidence for future national and

489 international regulations on air quality.



**Author contribution**
X.R., R.D., H.H. and Z.L. designed the aircraft campaign; H.H., X.R., F.W., X.D., and F.L. performed the research flights; Y.W., X.R., and T.Z. conducted the surface observations; H.H., T.P., and Y.H. developed the modeling system; H.H., X.R., and S.B. analyzed the data; H.H., X.R., S.B. and R.D wrote the paper

**Acknowledgements**
This work was funded by the National Science Foundation of the United States (Grant 1558259). We thank all of the A$^2$BC and ARIAs research team, especially the flight crew of Hebei Weather Modification Office's Y12 airplane. The flight campaign was conducted in association with the NASA's KORUS-AQ campaign.

**Disclaimer**
The scientific results and conclusions, as well as any views or opinions expressed herein, are those of the author(s) and do not necessarily reflect the views of NOAA or the Department of Commerce.

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





**Tables and Figures**

**Table 1.** List of CMAQ simulations with adjusted emissions based on satellite observations. Anthropogenic CO, $NO_x$, and VOCs emissions were adjusted using MOPITT CO, OMI $NO_2$, and OMI HCHO observations.

| Run NO. | Experiment Name | Bio. VOCs | Anthro. CO | Anthro. $NO_x$ | Anthro. VOCs |
|---|---|---|---|---|---|
| 1 | CMAQ_baseline | BEIS | EDGAR | EDGAR | EDGAR |
| 2 | CMAQ_noBEIS | N/A | EDGAR | EDGAR | EDGAR |
| 3 | CMAQ_CO | BEIS | Adjusted | EDGAR | EDGAR |
| 4 | CMAQ_$NO_x$ | BEIS | EDGAR | Adjusted | EDGAR |
| 5 | CMAQ_VOCs | BEIS | EDGAR | EDGAR | Adjusted |
| 6 | CMAQ_All | BEIS | Adjusted | Adjusted | Adjusted |



**Table 2.** Statistics of CMAQ performance of six sensitivity experiments compared with ARIAs
aircraft measurements over the NCP.

| No. | Name | Mean Diff | Slope | Stdev | Corr. R | NMB | NME | RMSE | Mean Ratio |
|---|---|---|---|---|---|---|---|---|---|
| | | ppbv | Unitless | ppbv | Unitless | % | % | ppbv | Unitless |
| | | | | | $O_3$ | | | | |
| 1 | CMAQ_Baseline | -21.35 | 0.56 | 13.25 | 0.37 | -25.14 | 25.86 | 25.10 | 0.75 |
| 2 | CMAQ_noBEIS | -23.74 | 0.49 | 12.85 | 0.41 | -27.94 | 28.34 | 26.97 | 0.72 |
| 3 | CMAQ_NOx | -19.83 | 0.59 | 13.63 | 0.34 | -23.34 | 24.18 | 24.03 | 0.77 |
| 4 | CMAQ_VOCs | -19.26 | 0.66 | 13.66 | 0.36 | -22.67 | 23.81 | 23.58 | 0.77 |
| 5 | CMAQ_CO | -20.35 | 0.61 | 13.52 | 0.36 | -23.96 | 24.83 | 24.40 | 0.76 |
| 6 | CMAQ_All | -15.18 | 0.81 | 14.83 | 0.33 | -17.87 | 20.33 | 21.18 | 0.82 |
| | | | | | CO | | | | |
| 1 | CMAQ_Baseline | -183.56 | 0.21 | 165.92 | 0.23 | -60.26 | 60.26 | 246.98 | 0.40 |
| 2 | CMAQ_noBEIS | -186.34 | 0.21 | 165.52 | 0.25 | -61.17 | 61.17 | 248.79 | 0.39 |
| 3 | CMAQ_NOx | -184.25 | 0.21 | 165.76 | 0.24 | -60.48 | 60.50 | 247.39 | 0.40 |
| 4 | CMAQ_VOCs | -181.89 | 0.22 | 166.32 | 0.22 | -59.71 | 59.78 | 246.01 | 0.40 |
| 5 | CMAQ_CO | -148.55 | 0.36 | 167.90 | 0.22 | -48.76 | 50.32 | 223.67 | 0.51 |
| 6 | CMAQ_All | -104.45 | 0.52 | 175.48 | 0.21 | -34.29 | 45.03 | 203.60 | 0.66 |
| | | | | | $NO_2$ | | | | |
| 1 | CMAQ_Baseline | -1.72 | 0.31 | 3.09 | 0.58 | -59.91 | 64.59 | 3.52 | 0.40 |
| 2 | CMAQ_noBEIS | -1.73 | 0.31 | 3.09 | 0.58 | -60.47 | 64.90 | 3.52 | 0.40 |
| 3 | CMAQ_NOx | -1.45 | 0.38 | 2.99 | 0.60 | -50.61 | 61.26 | 3.31 | 0.49 |
| 4 | CMAQ_VOCs | -1.76 | 0.31 | 3.10 | 0.58 | -61.60 | 65.66 | 3.55 | 0.38 |
| 5 | CMAQ_CO | -1.70 | 0.31 | 3.09 | 0.58 | -59.28 | 64.20 | 3.51 | 0.41 |
| 6 | CMAQ_All | -1.47 | 0.38 | 3.01 | 0.59 | -51.23 | 61.62 | 3.33 | 0.49 |
| | | | | | NO | | | | |
| 1 | CMAQ_Baseline | -0.25 | 0.99 | 0.47 | 0.68 | -32.23 | 45.4 | 0.53 | 0.68 |
| 2 | CMAQ_noBEIS | -0.24 | 1.02 | 0.48 | 0.68 | -31.09 | 45.66 | 0.54 | 0.69 |
| 3 | CMAQ_NOx | -0.08 | 1.31 | 0.59 | 0.67 | -9.75 | 50.01 | 0.59 | 0.90 |
| 4 | CMAQ_VOCs | -0.30 | 0.89 | 0.45 | 0.68 | -38.63 | 46.58 | 0.54 | 0.61 |
| 5 | CMAQ_CO | -0.26 | 0.96 | 0.47 | 0.68 | -33.08 | 45.26 | 0.53 | 0.67 |
| 6 | CMAQ_All | -0.16 | 1.13 | 0.52 | 0.67 | -20.31 | 45.46 | 0.54 | 0.80 |
| | | | | | $NO_y$ | | | | |
| 1 | CMAQ_Baseline | -15.26 | 0.30 | 10.15 | 0.39 | -77.58 | 77.58 | 18.27 | 0.22 |
| 2 | CMAQ_noBEIS | -15.50 | 0.29 | 10.15 | 0.40 | -78.81 | 78.81 | 18.47 | 0.21 |
| 3 | CMAQ_NOx | -14.24 | 0.37 | 10.20 | 0.37 | -72.39 | 72.39 | 17.46 | 0.28 |
| 4 | CMAQ_VOCs | -15.23 | 0.30 | 10.16 | 0.39 | -77.42 | 77.42 | 18.25 | 0.23 |
| 5 | CMAQ_CO | -15.26 | 0.30 | 10.15 | 0.39 | -77.56 | 77.56 | 18.27 | 0.22 |
| 6 | CMAQ_All | -14.21 | 0.37 | 10.20 | 0.37 | -72.26 | 72.26 | 17.44 | 0.28 |




**Figure 1.** ARIAs flights over the NCP and the WRF-CMAQ domains. Eleven Research Flights
(RF) were conducted in May to Mid-June 2016. CMAQ has two domains, the coarse domain
(d01, 36 km resolution) covering East Asia and the nested domain (d02, 12 km resolution)
focusing on eastern China. a) Summary of flight routes; b) WRF-CMAQ modeling domain (the
red dot represents the location of the Xingtai supersite).
a)
b)

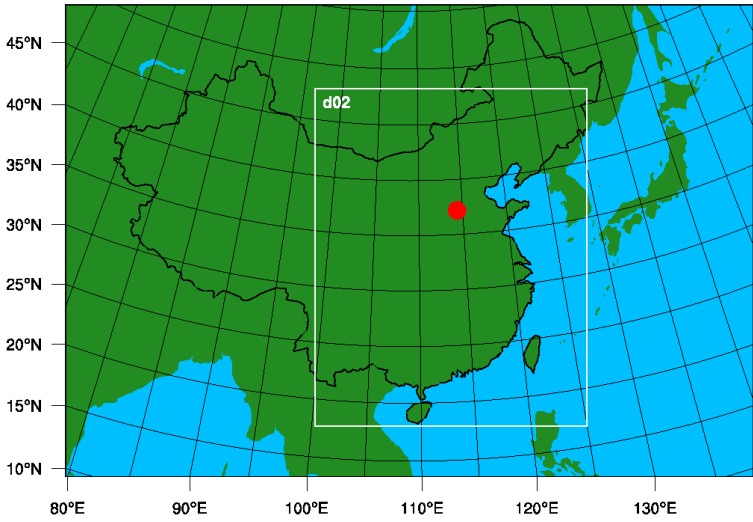




**Figure 2.** Summary of air pollutant concentrations in the NCP observed by Y12 aircraft. a) $O_3$, b)
$NO_2$, c) CO, and d) $CO_2$.

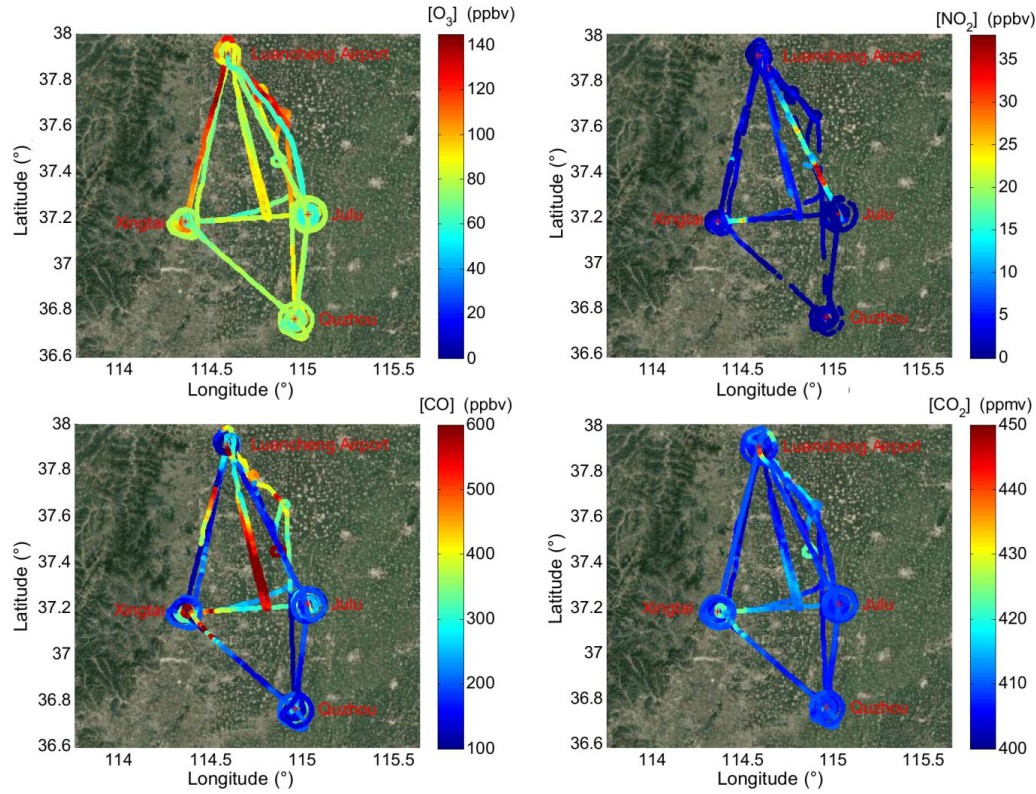




**Figure 3.** Vertical profiles of air pollutants over four locations in the NCP. a) Xingtai (XT), b) Luancheng (LC), c) Julu (JL), and d) Quzhou (QZ). Red lines show the mean profiles.

a)

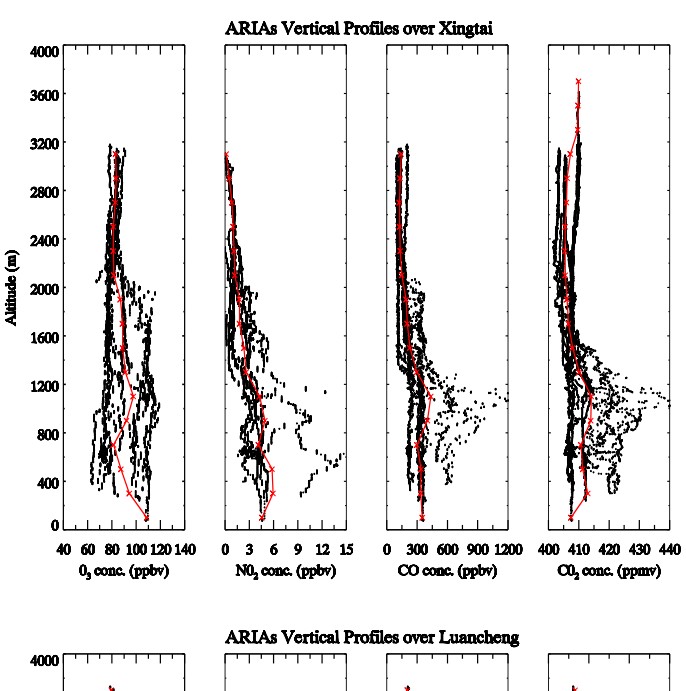

b)

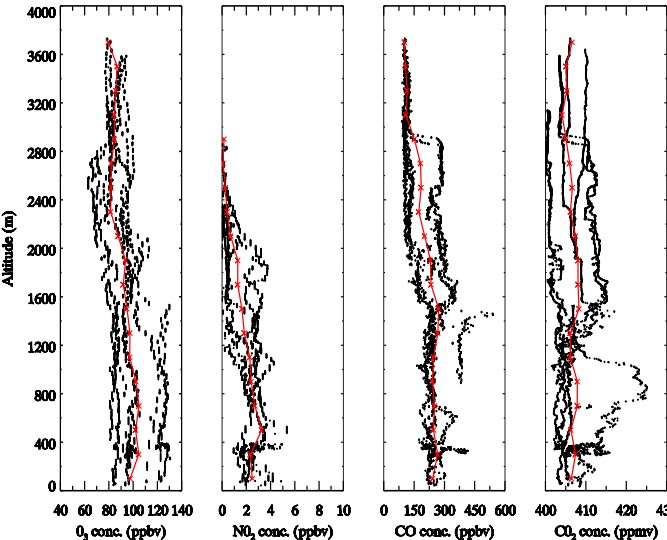





c)

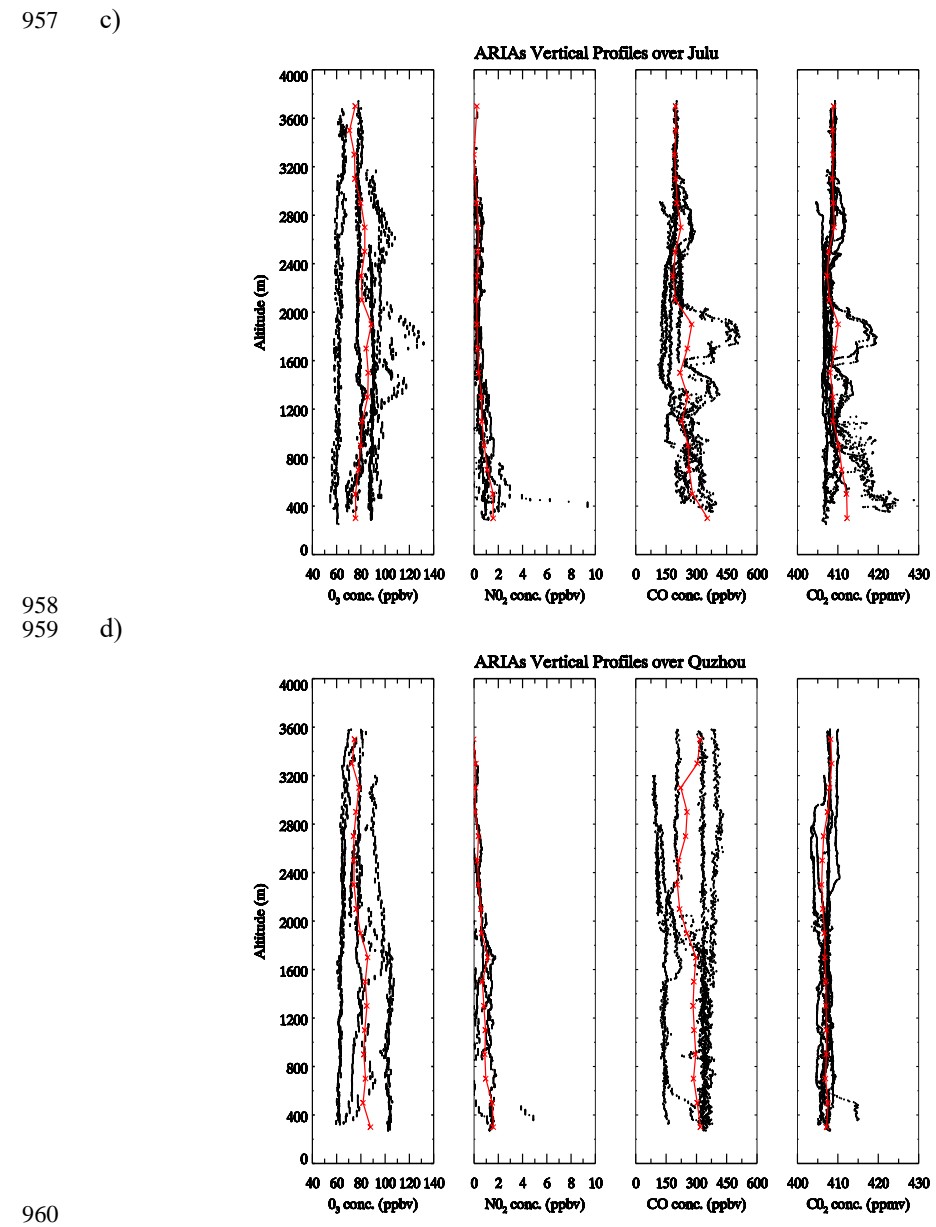

d)





**Figure 4.** Comparison of 10-min averaged aircraft data and CMAQ simulations from 11 ARIAs research flights. a) $O_3$, b) CO, c) NO, and d) $NO_2$. Black line shows the 1:1 ratio; red line stands for the linear regression fitting line. M_Diff: mean difference; R: correlation; NMB: normalized mean bias; NME: normalized mean error; RMSE: root-mean square error; M_Ratio: mean ratio.

a)

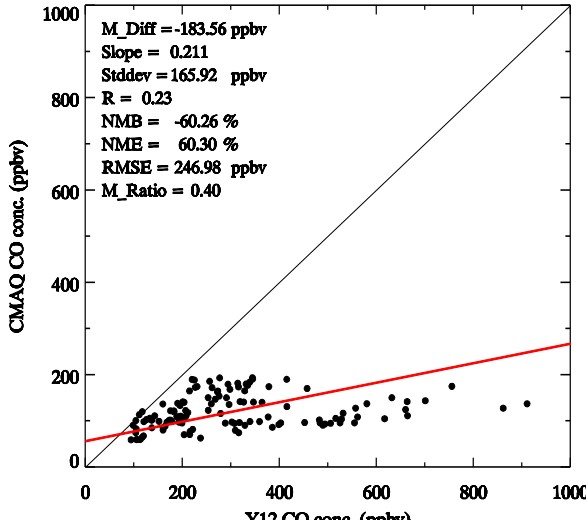


b)





c)

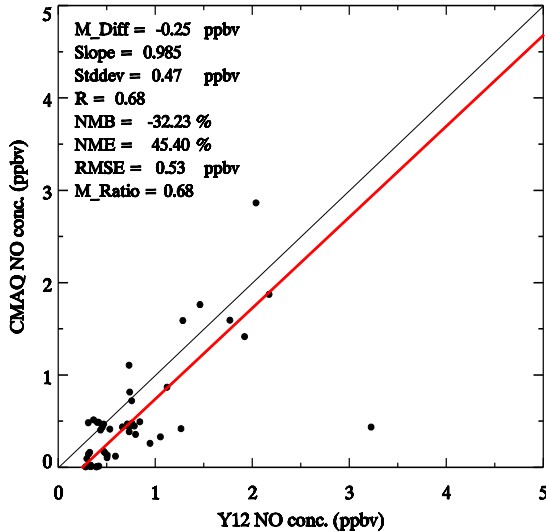


d)

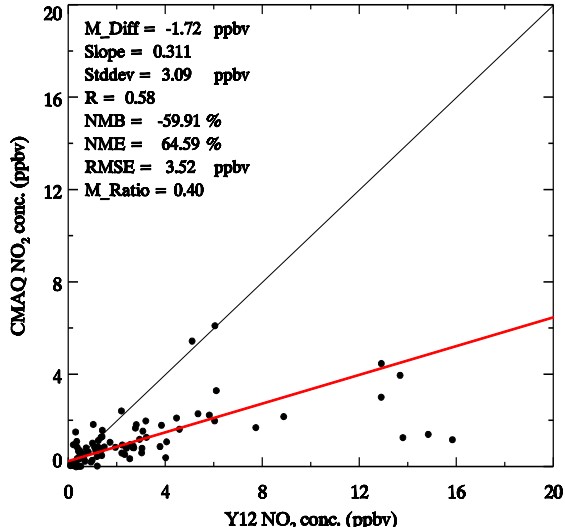




**Figure 5.** Comparison of total VOCs concentrations from WAS samples and CMAQ simulations.
Values are in unit of parts per billion Carbon (ppbC). Black line shows the 1:1 ratio; red line
stands for the linear regression fitting line. M_Diff: mean difference; R: correlation; NMB:
normalized mean bias; NME: normalized mean error; RMSE: root-mean square error; M_Ratio:
mean ratio.

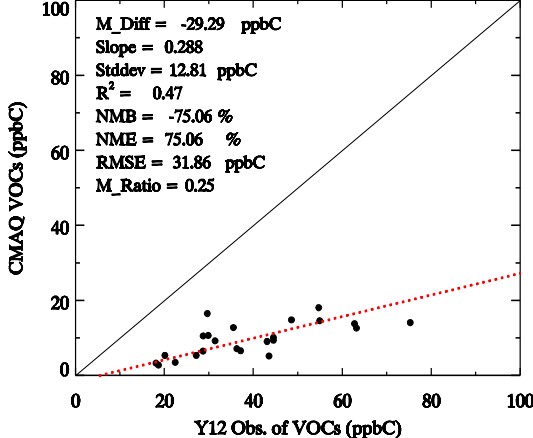




**Figure 6.** Comparison of emission enhancements (EEs) from the ARIAs campaign and emission
factors (EFs) from the EDGAR emission inventory. a) $\Delta CO/\Delta CO_2$, b) $\Delta NO_x/\Delta CO_2$, c)
$\Delta NO_x/\Delta CO$. Blue histogram shows the distribution of EEs observed by the Y12 aircraft; red line
shows the ratio calculated using the EDGAR anthropogenic emissions.
a)

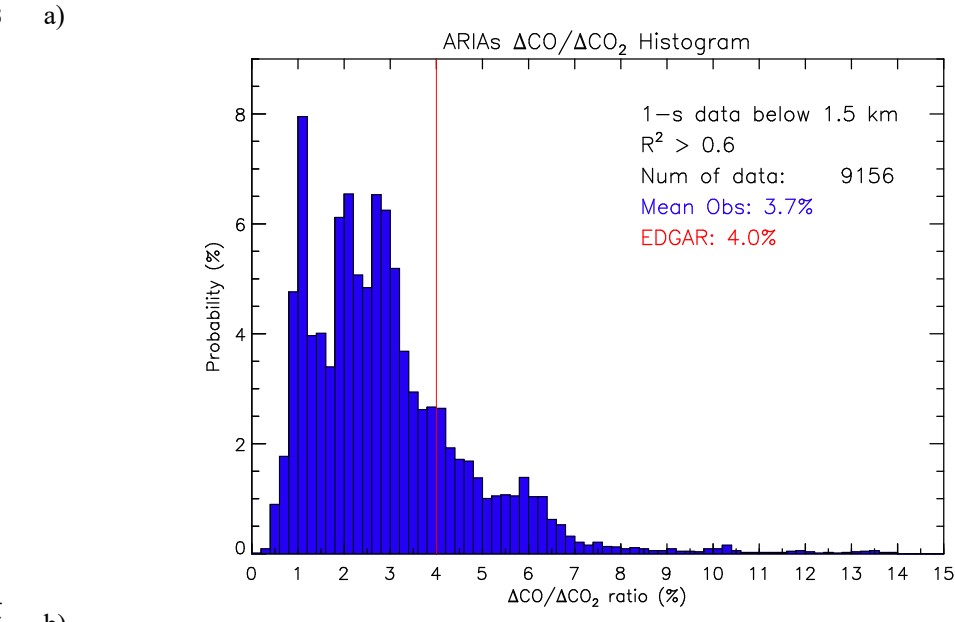

b)

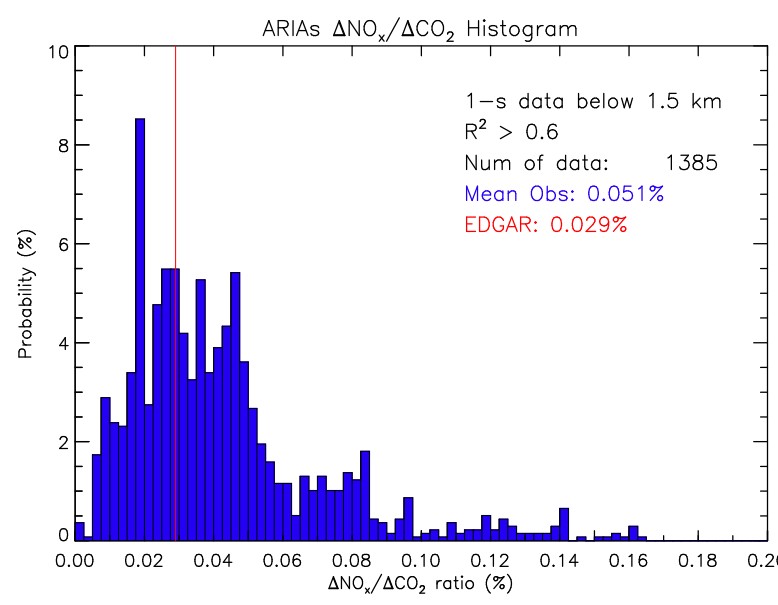




c)

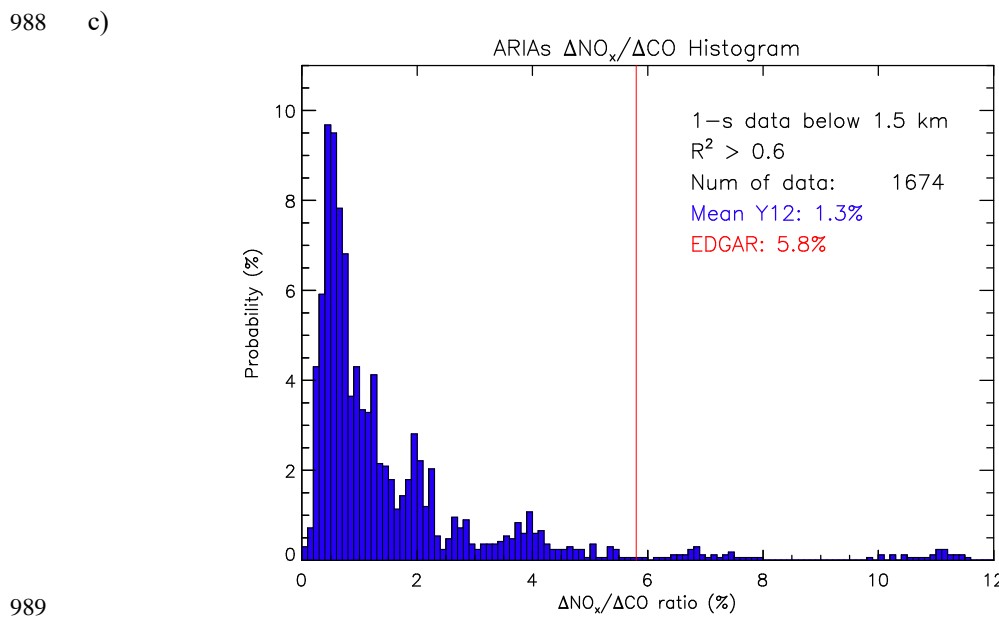






**Figure 7.** Comparison of air pollutants from satellite observations and CMAQ simulations. a) OMI $NO_2$ column (left) and the difference between OMI and CMAQ (right), Unit: Dobson Unit (1 DU = $2.69 \times 10^{20}$ molecules/cm$^2$); b) OMI HCHO column (left) and the difference between OMI and CMAQ (right), Unit: DU; c) MOPITT near surface CO (left) and the difference between MOPITT and CMAQ (right), Unit (ppbv).

a)

b)

c)





**Figure 8**. Ratios of column contents of the baseline CMAQ simulations and satellite
observations. a) CMAQ/OMI $NO_2$; b) CMAQ/OMI HCHO; c) CMAQ/MOPITT CO.
a)

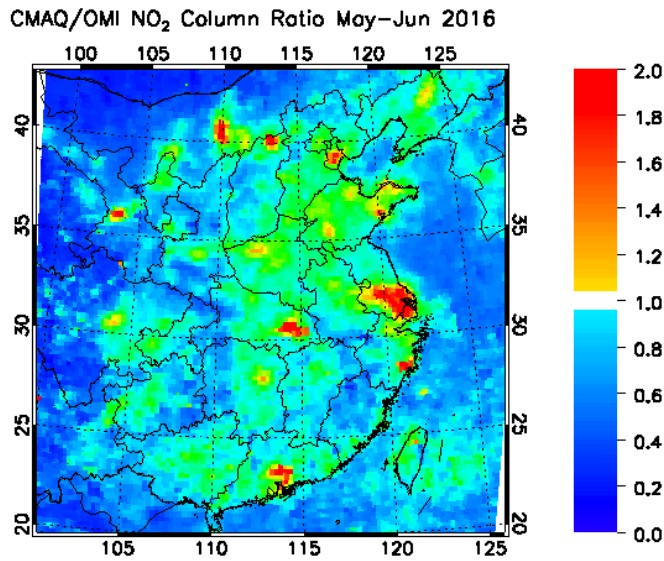


b)

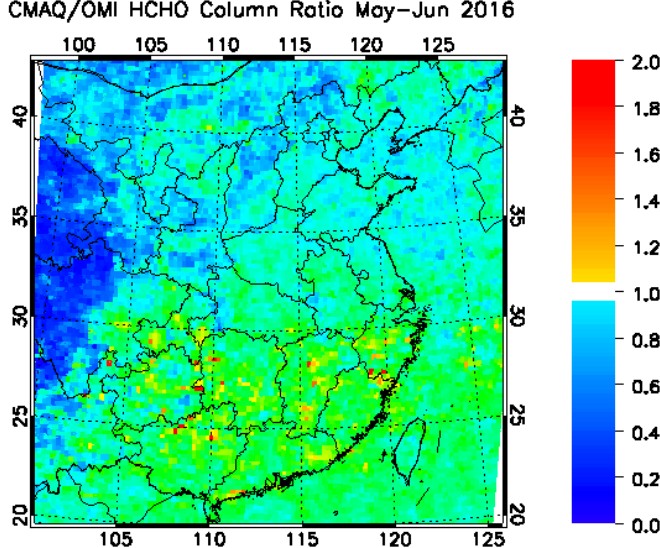




c)

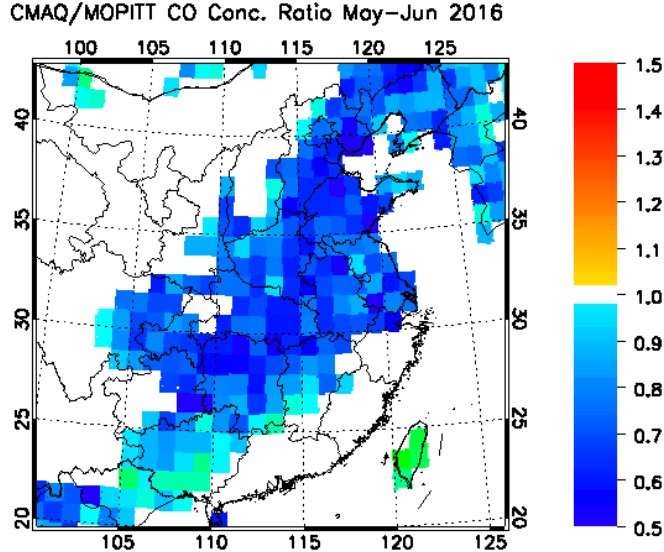






**Figure 9.** Column HCHO/NO$_2$ ratios over East Asia in spring 2016. a) Ratio derived from collocated OMI HCHO and NO$_2$ observation; b) Ratio calculated from CMAQ simulations with OMI quality information and averaging kernel (AK).

a)

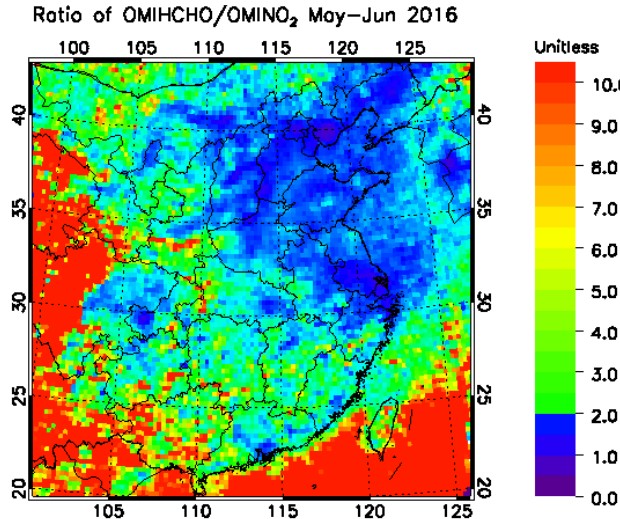

b)

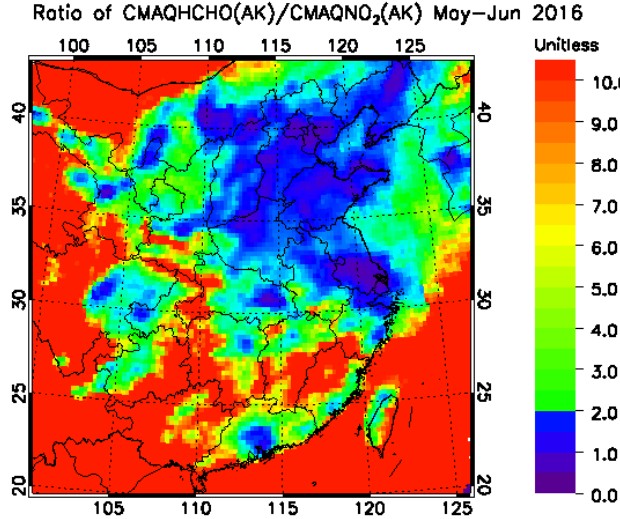



**Figure 10.** Comparison of surface hourly observations of air pollutants and CMAQ simulations
at the Xingtai supersite from May to mid-June 2016. a) $O_3$, b) CO, c) $NO_2{}^*$, d) $NO_x$, and e)
HCHO. *Surface $NO_2$ is inferred as $NO_x$-NO from surface observations.
a)
b)

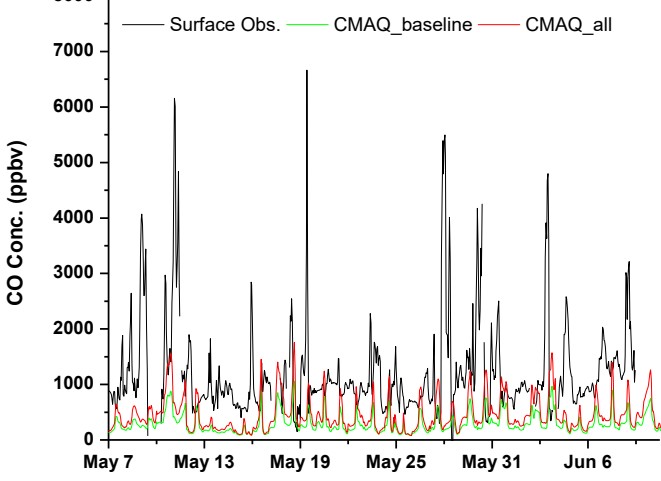




c)

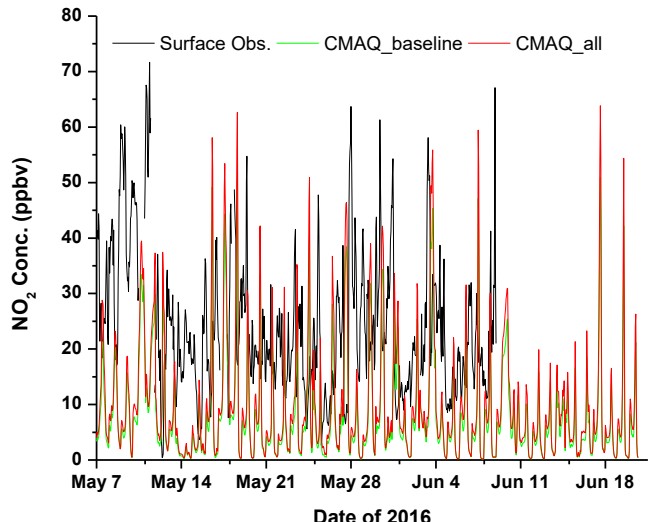

d)

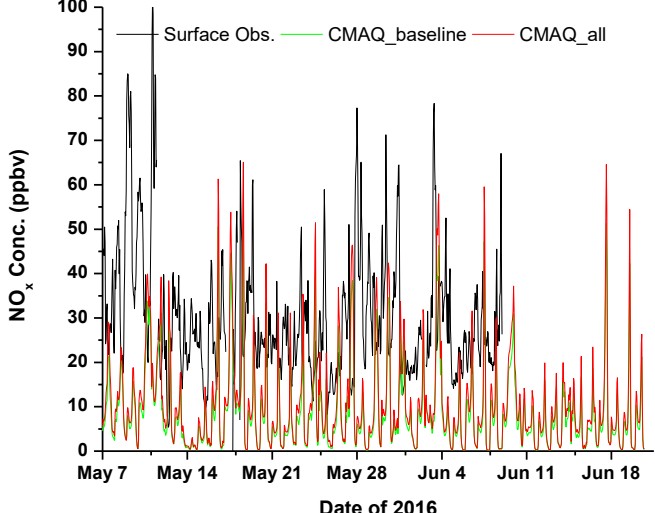




e)

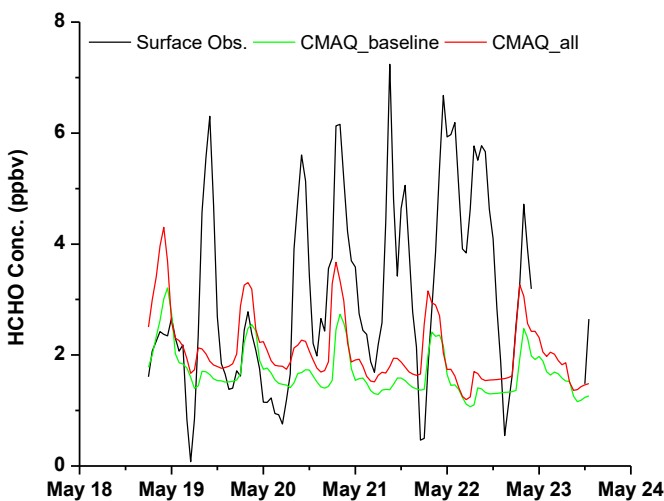






**Figure 11.** A case study comparing aircraft observations and the CMAQ_All case results on June
11, 2016. Background: CMAQ simulations. Overlay: 1 min Y12 measurements. a) $O_3$, b) CO, c)
$NO_2$, d) NO, and e) $NO_y$.
a)

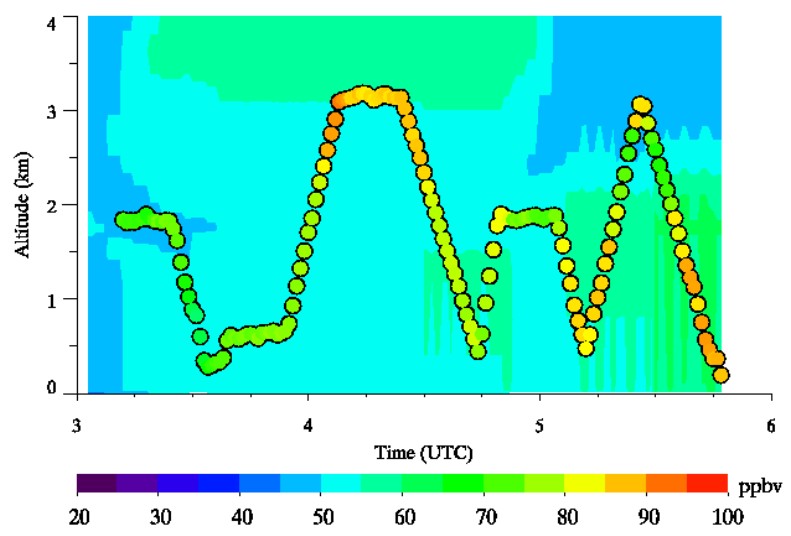

b)

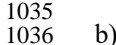

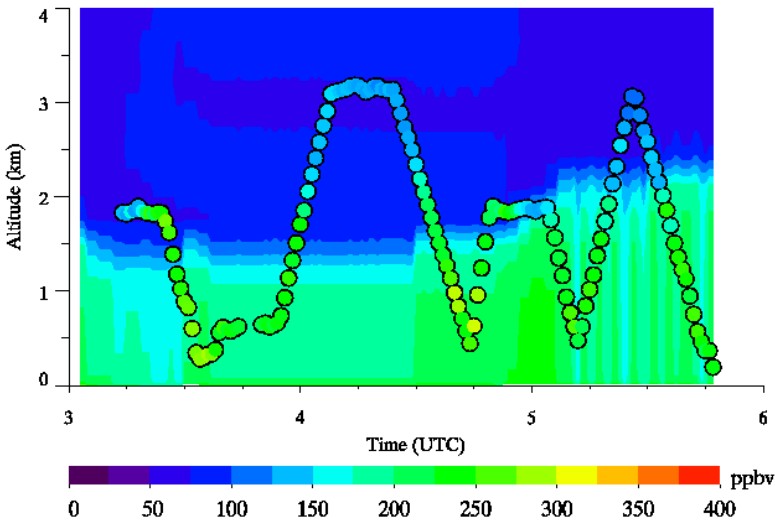




c)

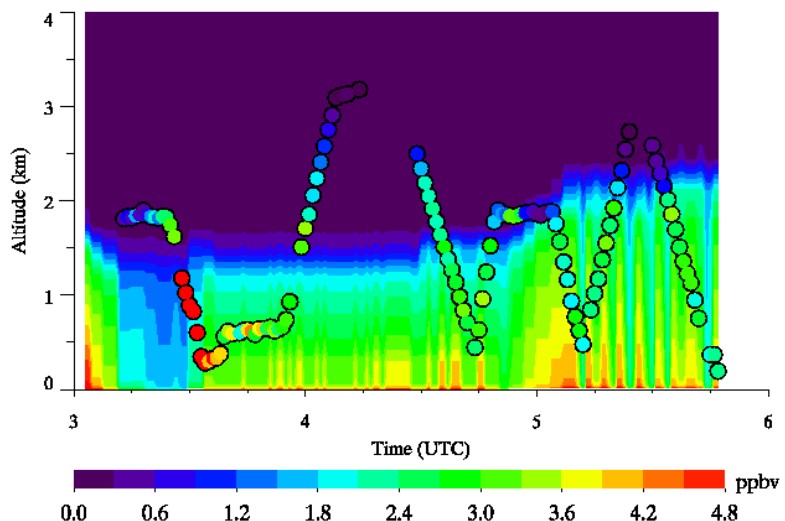

d)

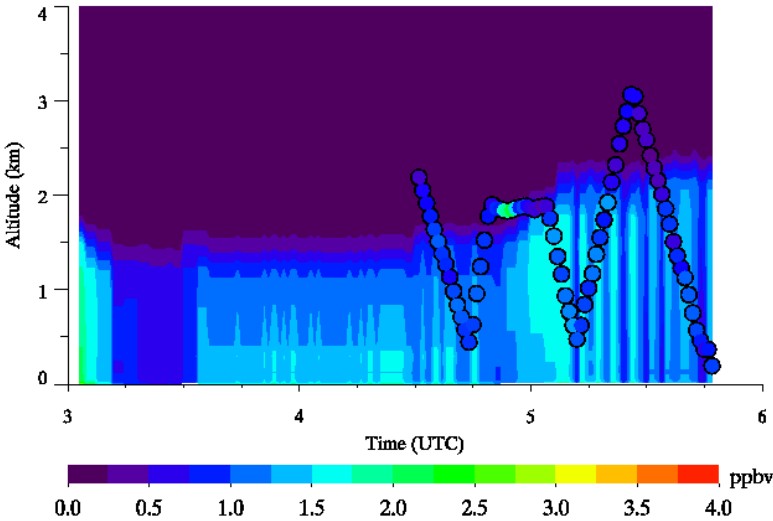






e)

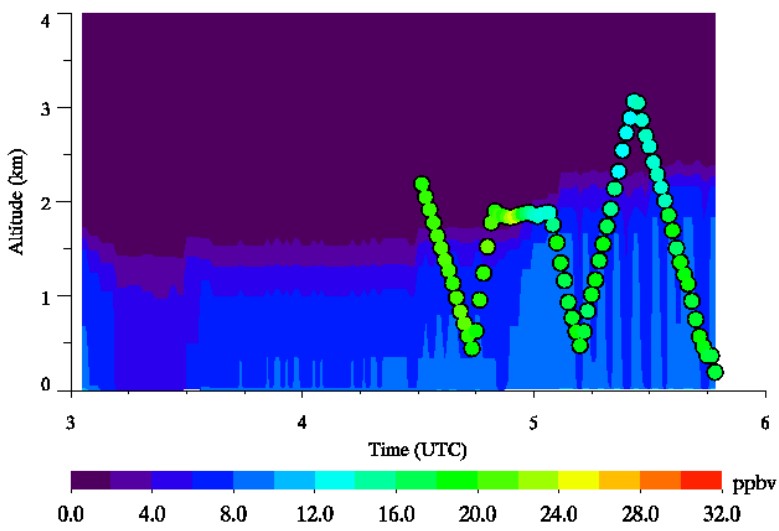
