# Peer review of "Manuscript under review for journal Atmos. Chem. Phys."

_Atmospheric Chemistry and Physics, 2019_

## Referee Comment (RC1) · Anonymous Referee #1 · 25 Apr 2019

Review of "Evaluation of Anthropogenic Emissions and Ozone Pollution in the North China Plain: Insights from the Air Chemistry Research in Asia (ARIAs) Campaign" by Hao He et al.

MS Number: acp-2019-248

**Summary:**

This paper discusses aircraft and surface observations of important air pollutant species over China. Such measurements are critically important for understanding the poor air quality in China, which adversely affects the health of 100's of millions of people. However, this paper does not adequately describe the measurements, and the results are poorly discussed, in some cases reaching contradictory conclusions. In at least one instance, the results are compromised by a fundamental math error. The results of the photochemical grid modeling appear to be extremely poor. I recommend that the paper be rejected, and the authors encouraged to resubmit the paper, if the issues detailed below can be adequately addressed.

**Major issues and comments:**

1) Much more detailed information regarding the aircraft measurements must be given, perhaps in the Supplementary Material. The single paragraph in Section 2.1 is inadequate. Questions that immediately occurred to me include: Were the instruments zeroed and calibrated in flight? How can ozone be measured at 1 Hz frequency when the specifications of the ozone instrument state that the response time is 20 seconds with a 10 second lag time? How can formaldehyde be measured at 1 Hz frequency when the specifications of that instrument state that the time resolution is 90 sec with a delay time ~300 sec. How were the lag times of the various instruments synchronized with the GPS system? What was the aircraft air speed (i.e., 1 Hz measurement frequency corresponds to what spatial resolution)? What was the rate of ascent and descent (this is important given the evident time resolution of some of the instruments)? Given the disparate time resolution of the instruments, how were comparable spatial average concentrations calculated? What are the accuracies and precisions of the 1 Hz measurements? (Please provide a table listing these instruments parameters and explanations or references of how these parameters were determined.) Can any evidence be provided to demonstrate these tabulated accuracy and precision parameters are realistic (e.g., the results of in-flight instrument comparisons with another aircraft)? A full description of these and all such instrument issues must be provided for the interested reader.

2) Lines 264-276: This paragraph suggests that underestimation of ozone precursors in CMAQ could lead to the poor model performance. Poor simulation of the boundary layer depth also could lead to the poor model performance; a balanced discussion of both of these issues should be given.

3) Figure S2d seems to show a comparison of measured versus modeled NO and NOy, yet the measurement of these two species is not described in Section 2.1. What is going on here? Similarly Figure 4c gives NO measurements. If NOy was measured, why is its model-measurement comparison not given in Figure 4? Clearly the description of the aircraft measurements must be improved as noted in Comment 1 above.

4) Figure 4 compares 10-minute averages of measured concentrations with model results. A discussion of this averaging must be given, as the aircraft covers a significant distance (~30 km?) and can cover a significant altitude range in 10 minutes. Is this really a reasonable comparison?

5) Lines 264-276: This paragraph is confused, contradictory and inaccurate. Figure 4 shows that NO is neither under- nor over-estimated by CMAQ. Figure S2 indicates that CMAQ under-estimates (not over-estimates) NOy. This description requires rethinking and rewriting.

6) It is not possible for me to assess the validity of the analysis in Section 3.2 because 1) only a very brief discussion of the approach and results are given, 2) the reference for the method (Halliday et al., 2018) is "to be submitted" and is thus not available, and 3) no illustrations of the EF calculations are shown in the paper or in the Supplementary Material. The description of this analysis must be improved.

7) Lines 307-328: In these paragraphs the authors apparently make an elementary math error. They appear to be comparing the average of many, independently determined emission ratios (EFs) based on measurements with the ratio of the total EDGAR inventory emissions. Such a comparison is not valid because in general the arithmetic average of a distribution of ratios is not equal to the ratio of the means of the numerators and denominators of the ratios. The underlying numbers must be corrected and the discussion modified accordingly.

8) Section 3.3 compares satellite measurements with CMAQ model results for $NO_2$, $CH_2O$ and CO. The discussion proceeds with no considerations of systematic uncertainties in either the satellite measurements or the model results. The authors note that over the aircraft campaign area CMAQ predicts 81% of $NO_2$ satellite column measurement and has good agreement with $CH_2O$ (<20% underestimation). To me this seems to be excellent agreement, and that it is not legitimate to interpret <20% differences as indications of emission uncertainties. This section is not acceptable without robust quantitative discussion of the systematic uncertainties in both the satellite measurements and the model results.

9) Sections 3.1 and 3.3 seem to reach inconsistent conclusions. Section 3.1 suggests that CMAQ generally underestimates observed concentrations of major air pollutants, often by large factors (Line 273: factors of 2 to 4 for all air pollutants; Line 287: a factor of 5 for VOCs). Yet as noted in the previous comment, Section 3.3 finds agreement within 20%. Section 3.4 goes back to the idea that the CMAQ run substantially underestimates the concentrations of ozone and its major precursors in the NCP. Such inconsistencies must be fully and quantitatively addressed before emissions within the CMAQ modeling can be objectively adjusted.

10) Section 3.4 is not satisfactory. Figures 10 and S3 present time series of observations and model results, but the agreement is quite poor regardless of the model run. These comparisons should be based on an objective measure of overall model performance so that the reader can appreciate how well or poorly each of the model simulations actually reproduced the observations.

11) Figure 11 compares observations and model result for a selected flight segment. The agreement appears to be extremely poor. Again, these comparisons should be based on an objective measure of overall model performance (for all flights) so that the reader can appreciate how well or poorly the model simulation actually reproduced the observations.

12) Much of the Conclusions and Discussion section is speculative and/or not quantitatively supported by the results discussed previously.

**Minor issues:**

1) Line 122: The country of the Environmental Protection Agency (EPA) should be indicated.

China (where the research is located) has an EPA, but I assume that this sentence refers to the U.S. EPA.

2) Figure 1 should be improved. It is not possible for a reader unfamiliar with Chinese geography to easily understand the region of China actually covered by the flight tracks.

3) Lines 245-248: The statement on these lines is not accurate. The description is not of the generally observed concentrations, but rather reflects the maximum concentrations observed.

4) The Sections are not properly numbered; two are labeled Section 3.3.

---

## Referee Comment (RC2) · Anonymous Referee #2 · 18 Jun 2019

This manuscript focuses on the comparison of ground/airborne measurements and satellite retrievals of a variety of air quality relevant trace gases with CMAQ model products. The authors attempt to adjust emissions in CMAQ based on column model/satellite ratios, thus improving the modelled ozone agreement with ground/airborne measurements. The dataset described in the manuscript is certainly novel, but there are enough omissions in the analysis that would prohibit its publication in its current state. I recommend the manuscript to be rejected, while encouraging the authors to resubmit after the address of the below issues.

[Figure]

**[ACPD](javascript:void(0))**

Interactive
comment

Major comments:

-The improvement in ozone agreement is quite modest, to the point I am interested as to how significant it is. How does this compare with the various uncertainties in measurements and the EI? What ratios would be required to force greater agreement? It seems there would be many more sensitivity studies needed to try to evaluate this. As is, this suggests to me that the ozone discrepancy cannot be explained by a simpler large scale emissions adjustment and resolving it would require a more intensive study of the VOC chemistry.

-A prevalent assertion of the authors is that discrepancies between model and observations are driven by reduced emissions, but the authors do not address any validation of the 2010 emissions. It is difficult to claim difference is due to emissions reduction without evidence that CMAQ and observations, airborne or satellite, agreed in the first place. Consistent, yes, but not necessarily suggestive or conclusive. The authors do not cite other studies to support this. Language should be adjusted throughout the manuscript to reflect that the adjustments needed to improve the model agreement are inherent, a component of which is consistent with emissions reduction.

-Line 296-306: the EF technique is not very well described here. How were the cutoffs that the authors state ($R^2$, altitude, etc) justified? A fuller description of the technique would help significantly.

-Line 296-306: Since these come a variety of emission sources, a simple mean seems simplistic to describe the population. In looking at Fig 6, there seem to be multiple overlapping populations contributing to the mean. A broader discussion of these populations seems warranted.

-Line 277-293: Where do the Y12 NOy measurements come from? There is no description in the instrument section. Without the details of the measurement, this entire discussion of reactive nitrogen is not possible to evaluate.

[Printer-friendly version](javascript:void(0))

[Discussion paper](javascript:void(0))

-Sect 3.3: It seems there is should be a before/after version of this analysis: one with the baseline CMAQ model (which is to my understanding what is used for this analysis) and with at least the CMAQ_all case. That seemed to be where the analysis was heading, and without it, the manuscript feels incomplete.

Minor comments:

-Fig 1a probably unnecessary, label fig 2 more clearly

-Fig 3: change zeros to O

-Fig S1 & S3: these have very similar captions, though the difference is clearer from the manuscript. I would edit the captions to highlight the differences

-Line 251, "We observed isolated plumes...": Where are these plumes with respect to the surface layer structure? What about the secondary plumes at 800-1200 m? The authors need to add context on the atmospheric structure to describe the transport impact of these plumes. Perhaps a summary plot of potential temperature as well?

-Line 259, "In summary...": This sentence was somewhat confusing, is unclear whether authors mean exclusively east-west gradients observed or both north-south and east-west.

-Fig 7: highlight here the aircraft campaign area, is difficult to eyeball where it is using fig 1b

-Fig 7 & 8: having both the differences and ratios seems redundant. As ratios are used in the later analysis, the authors should keep those figures and get rid of the difference figures, with some minor rewording in section 3.2.

-Fig. 9: it is not clear to me from the discussion the need for both panels A & B

-Fig 10: these are difficult to evaluate with the high variability. Perhaps a difference plot between observed and modeled, or ratios of the same, in order to clarify the improvement

[Figure]

---

## Author Comment (AC1) · 19 Aug 2019

Please see the attached PDF files for our response and the revised manuscript.

Please also note the supplement to this comment:
https://www.atmos-chem-phys-discuss.net/acp-2019-248/acp-2019-248-AC1-supplement.zip

―――――――――――――――――――――

2019.